# The Influence of Alkali Content on the Hydration of the Slag-Based Geopolymer: Relationships between Resistivity, Setting, and Strength Development

**DOI:** 10.3390/polym15030518

**Published:** 2023-01-18

**Authors:** Yuan Fang, Kunde Zhuang, Dapeng Zheng, Weitao Guo

**Affiliations:** Key Laboratory for Resilient Infrastructures of Coastal Cities (MOE), College of Civil and Transportation Engineering, Shenzhen University, Shenzhen 518060, China

**Keywords:** slag-based geopolymer, electrodeless resistivity, non-destructive monitoring, alkali content, hydration process

## Abstract

This study investigated the influence of alkali content on the early-age hydration process of slag-based geopolymer and the feasibility of non-destructive resistivity. Results showed that there existed a threshold of alkali content in adjusting the early-age hydration. Initially, increasing the alkali content tended to accelerate the dissolution period (detected by resistivity and heat release rate) and resulted in a denser microstructure (detected by TEM). When the alkali content surpassed 6 wt%, the increasing alkali content mitigated the structural development of a slag-based geopolymer, as it lowered the liquid water content and caused local precipitation, which trapped the early-age ions transmission and, therefore, the later-age mechanical development was inhibited. It was proven that the resistivity acted as a linkage among the reaction degree, workability, and strength development.

## 1. Introduction

The increasingly serious environmental issues have promoted research in regard to novel construction materials. Due to the massive utilization of industrial wastes [1,2], superior properties in strength and durability [3,4], and fewer CO_2_ emissions compared with ordinary Portland cement (OPC) [5,6,7], geopolymer comes to the front.

The term ‘geopolymer’ refers to the hardening product of the reaction between aluminosilicate sources and alkaline activators [8], while those geopolymeric gels derived from mixtures of slag and activators are also defined as ‘alkali-activated materials’ [9]. There is a consensus that during the early-age reaction of geopolymers, the Si–O–Al bonds in the aluminosilicate sources are broken and reconstructed under the catalysis of hydroxyl released by the alkaline activator, accompanied by the stepwise forming of the initial complex, the intermediate gelling phase, and the final framework of layered or network C–(N,K)–A–S–H(CaO–(Na_2_O,K_2_O)–Al_2_O_3_–SiO_2_–H_2_O) gel, the main strength-giving product in geopolymers [10]. The intrinsic properties of geopolymers are governed by various factors, including compositions of aluminosilicate sources, water-to-solids ratio, as well as the type and dosage of the activator. Typically, the Ca/Si ratio of the raw materials determines the basic kind of hydration product (for the high-calcium system, the hydration product is a layered C–A–S–H; for the low-calcium system, the hydration product is designated as a network (N,K)–A–S–H [10]), while the other factors (for instance, alkali content) mainly affected the reactant concentration and the hydration rate, resulting in variations of crystallinity [11], workability [12], total porosity [13], etc.

Viewing from a larger scale, the hardening of geopolymer paste is somewhat similar to that of OPC, both of which require the dissolution of the raw materials and precipitation of the hydration products on the unhydrated particles, followed by phase boundary contact and reduced pore volumes of the geopolymers pastes [14]. In this process, owing to the consuming of reactant and traps of ions in pore spaces in the solidified hydration products, the early-age reactions of both geopolymers and OPC are apparently divided into four stages from the aspect of the reaction rate, namely the wetting, acceleration, deceleration, and steady periods [15,16,17,18,19], as identified by the heat release rate. Generally, this early-age heat releasing is completed within 24 h, which has been proven to relate to the setting behaviors and later-age strength developments (3 days~28 days) of OPC and geopolymers. Although the four-stage reaction periods and gel phase formation for geopolymers have been discussed by recording the heat release, phase transformation, and micro-morphology, there is a need to correlate the reaction degree with the workability and strength development.

During the early-age hydration, the solid phases and pore solution with various ionic concentrations, which possess different electrical conductivity, are undergoing dynamic equilibrium, and thus the resistivity could be used to monitor the hydration process. The in situ electrodeless resistivity measurement has been successfully applied to OPC [14]. It was found that the inflection points on the resistivity curve were quantitatively (either a logarithmic or linear equation) related to the heat release, shrinkage, setting time, and strength development.

Even with the above successes of adopting the non-contact electrical resistivity method to monitor the hydration process of OPC, this method has not been applied in research on geopolymers with varying alkali contents. Therefore, it is highly significant to apply this method to analyze the process of the hardening reaction of geopolymers and to predict their early properties. In this study, the non-contact electrical resistivity method will be applied to the slag-based geopolymer activated by sodium silicate, and the effect of sodium silicate content on the hydration process and properties of the slag-based geopolymer will be explored. In addition, FTIR (Fourier-transform infrared spectroscopy) and SEM (search engine marketing) testing methods will be combined to assist in the analysis of the principle of slag-based geopolymer performance change. The needle penetration test and reaction heat test method will be used to verify the application of the non-contact electrical resistivity method in the slag-based geopolymer, and the fitting relation between the characteristic point of resistivity and the setting time will be obtained. This study introduces new research ideas on the influence of different alkali contents in the hardening reactions and strength development of slag-based geopolymers.

## 2. Experimental Program

### 2.1. Materials and Mix Proportions of Mortars and Pastes

The raw materials used for synthesizing slag-based geopolymer pastes are slag (BFS, manufactured by Shaoguan Iron & Steel Group Co. Ltd., Shaoguan, China), sodium silicate (at a SiO_2_/Na_2_O molar ratio of 2.0, produced in Guandong Province, China), and deionized water. X-ray fluorescence spectrometry (XRF) was applied to obtain the chemical composition of the slag (see Table 1). The particle size distribution of slag was measured by a DelsaMax CORE nanolaser particle size analyzer (Beckman Coulter, Shanghai, China); see Figure 1. The particle size of the slag ranged from 100 to 300 nm. The water-to-solid ratio was 0.35 and alkali content was 4–8 wt% (with an interval of 1 wt%). In this study, the alkali content is defined as the percentage of the mass of Na_2_O (g) in sodium silicate per unit of slag (g). For simplification, the samples are named in the form of SLX%, where X denotes the alkali content (for instance, SL5%).

For the mixing procedure, the sodium silicate and deionized water were mixed before adding to the slag powder, followed by 4 min of stirring (technic details are referred to as ASTM C305-12 [20]). Then, the mixture was immediately cast in a steel mold with dimensions of 40 mm × 40 mm × 160 mm and 40 mm × 40 mm × 40 mm. The specimens were stripped after 24 h and cured in a moist room (100% relative humidity and 23 ± 0.5 °C) for a specified number of days (7, 14, and 28 days). Three samples of each type were subjected to flexural and compressive strength tests.

### 2.2. Testing Methods

Samples for microscopic tests were prepared at the same mix proportions and curing conditions as those for mechanical performance tests. The fresh paste was used to test the fluidity and initial and final setting times. The test for spread fluidity of the freshly mixed paste was performed according to the GBT standard 2419-2005 [21] (i.e., the test method for the fluidity of cement mortar); initial and final setting times were measured using the Vicat test, according to ASTM C191 [22]. The viscosity of the freshly mixed paste was measured using an NDJ-79 rotary viscometer (Xuancheng, Shanghai, China), according to standard operating procedures.

The exothermic hydration rate of the groups (SL4~SL8%) was measured by a ToniCAL calorimeter (Toni Technik, Berlin, Germany) to assess the effect of alkali content on the hydration of the paste. The exothermic hydration rate was tested within 24 h. An electrodeless resistivity device (CCR-III, Beijing Radio Instrument No. 2 Factory, Beijing, China) was used for electrodeless resistivity measurement by grouting the fresh paste into the annular mold of the device and smoothing the paste surface using slight vibration to eliminate bubbles generated in the mixing process. Then, the mold was covered with a plastic film to reduce evaporation. The recording system automatically sampled at a threading interval of 5 s and resistivity was recorded for one day under ambient temperature maintained at 20 °C.

Fourier-transform infrared spectroscopy (FTIR-650S, TIANJIN GANGDONG SCI.&TECH. Co., Ltd., Tianjin, China) was used to determine the correlation between vibrational spectra and the reaction product formed in the slag-based geopolymer (SL4~SL8%). Samples for FTIR were prepared by mixing approximately 1 mg samples, obtained from carefully sealed specimens, with 300 mg of KBr. The wavenumber range was set between 4000–400 cm^−1^, with a resolution of 4 cm^−1^ and 64 times of repeat scanning.

Scanning electron microscopy (SEM, TM 250FEG, Quanta, Houston, TX, USA) was used to observe the morphology of the hydration products. Small fractured samples representing different hydration stages were dried at 50 °C for 24 h in a vacuum environment. The samples were then coated with 20 nm of gold to make them conductive. SEM was using a Quanta TM 250FEG instrument; the accelerating voltage of the SEM/energy-dispersive X-ray spectroscope (EDS) was maintained at 20 kV.

## 3. Results and Discussion

### 3.1. Resistivity Evolution of the Slag-Based Geopolymer with Various Alkali Contents

#### 3.1.1. Physical Meaning of the Resistivity Curve

The electrodeless resistivity curves (Figure 2) and differential curves (Figure 3) clearly indicated the periodical change of the paste, in which five stages could be observed (located between O to A, A to B, B to C, and after C). The physical meanings of these inflection points could be pre-summarized based on our previous research [15], although the detailed profiles were different due to the change of alkali content, and further discussion on the discrepancy is carried out in the following sections:Stage I (OA): the sodium silicate activates the slag and causes it to disintegrate. Under the attack of OH^−^, the slag bond break and decompose into SiO_4_^4−^, AlO_4_^5−^, and Ca^2+^, and simultaneously the initial complex precipitate near the surface of the slag particles. The inclined resistivity is mainly due to the rapid release of OH^−^;Stage II (AB): compared with Period I, the bulk resistivity starts to dominate by the precipitated hydrate phase. The latter causes a reduced pore volume, and thus, the conductivity declines. Although the reaction rate is still increasing, the slope of the differential curve shows a decline due to the boundary contact among slag particles (with the hydration layer attached to the surface);Stage III (BC): the first inflection (point B) denotes the decline of reaction rate, after which the hydrate layer is thicker enough to prevent the ionic migration from the unhydrated slag particles, which is named the induction stage;Stage IV&V (after C): although not obvious as the preceding stages, there exist acceleration and deceleration stages after point C (especially for the specimen with an alkali content of 9 wt%, see the green line in Figure 3), in which the former one (named the acceleration period) is due to the break of hydrate layer once the difference of osmotic pressure between the inner and outer part reached the threshold, releasing the residual ions from the unhydrated cores. After the residual ions are consumed, the total porosity further decreases and the reaction process enters the deceleration period.

**Figure 2 polymers-15-00518-f002:**
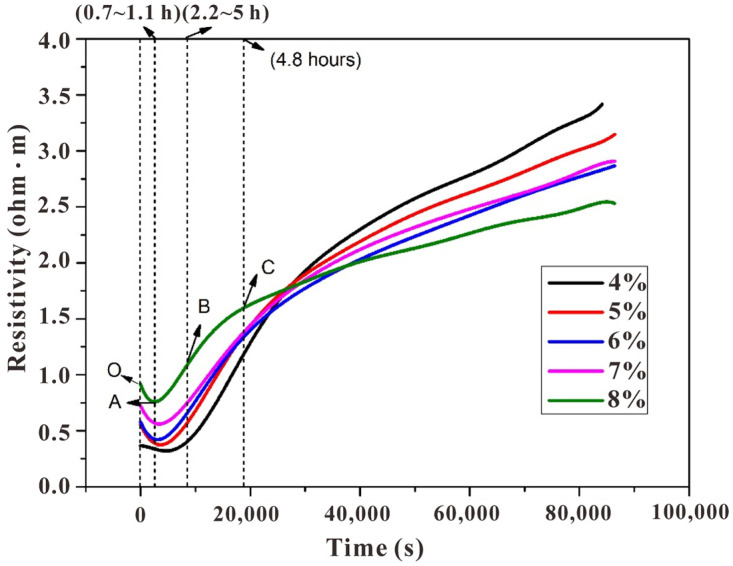
Electrodeless resistivity curves of a slag-based geopolymer with different alkali contents.

**Figure 3 polymers-15-00518-f003:**
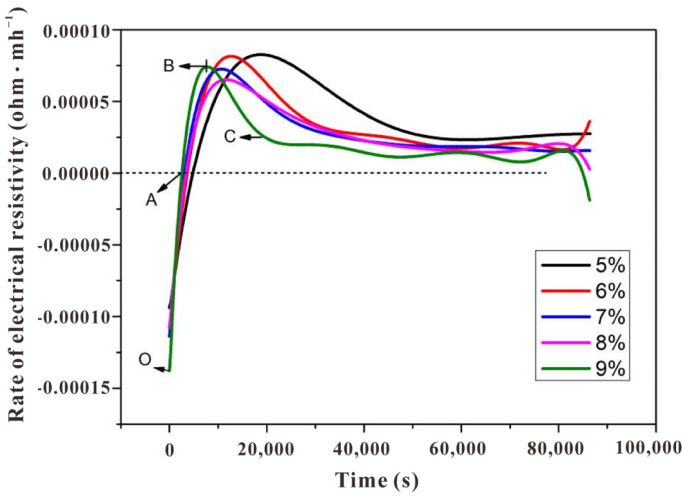
Differential electrodeless resistivity curves of a slag-based geopolymer with different alkali contents.

#### 3.1.2. The Effect of Alkali Content on the Resistivity Curves

The detailed profiles of resistivity curves were different from the cases in varying water-to-solids ratios [15], implying the significant impact of alkali content on the hydration process of slag-based geopolymer pastes, especially in stages I/II/III/IV. The resistivity curves in Figure 2 indicated that the higher the alkali content, the shorter the time needed before the resistivity reaches the minimum point A, and the greater the A point value. The time consumption was reduced from 1.6 h (SL4%) to 50 min (SL8%). This trend might have occurred because, in the dissolution stage (OA), a higher alkali content led to the faster formation of a layer on the surface of the slag. While increasing the alkali content advanced stage I, it also brought a shorter stage II, as depicted by the different positions of point B in Figure 3. Moreover, viewing from the profiles of the first peak in differential curves (Figure 3), it could be clearly observed that while the increasing alkali content accelerates the initial stages (see the advanced peak position), it caused the reaction system to enter the steady period much earlier (see the narrower peak width). Conclusively, although the sample with a low alkali content (4 wt%) exhibited a low reaction rate in the beginning, it seems that the reaction degree of it surpassed those with a higher alkali content (5~8 wt%) in the later age. This is also supported by the inverse relationship between resistivity and alkali content in Figure 2.

By combining with the resistivity model postulated in our previous study [15], the kinetic factor behind the change of curves in Figure 2 and Figure 3 could be preliminarily explained based on the model in Figure 4. On the one hand, the high alkali content increased the concentration of Na^+^ and SiO_4_^4−^; on the other hand, to keep the total water-to-solids ratio unchanged, increasing the alkali content (namely a higher dosage of a water glass) reduced the additional water. These two factors lead to (i) local precipitation of the gelling phase (described as the extra product in Figure 4) in the pore solution in addition to on the surface of the slag particles, which will not happen for those with a low alkali content (4 wt%), or the complex is not mature enough to trap the ionic transmission and (ii) weaker initial conductivity (or to say higher resistivity) due to the smaller amount of free water content. The reduced free water also mitigated the ionic transmission, and consequently, the reactant ions (OH^−^, SiO_4_^4−^, Na^+^) were unable to transmit to the unhydrated region (depicted by the enlarged region in Figure 4), leading to heterogeneous hydration for those pastes with high alkali content.

Due to the wide difference in electrical conductivity between the hardening product and pore solution, a higher resistivity value generally implies a higher reaction degree and superior strength, at least in the case of OPC or a slag-based geopolymer with the same alkali content [14,15]. However, it would be hasty to draw this conclusion herein, as the increasing alkali content complicated the situation. Firstly, it is still unable to estimate whether or not the contribution of liquid resistivity could cover the contribution from the part of the solid phase, resulting in the lower bulk resistivity for slag-based geopolymer pastes with higher alkali content (especially for sample SL5% and SL6%; there may exist a threshold of whether increasing alkali content would play a positive or negative role to the reaction degree and strength development). Secondly, the values of fluidity, shear viscosity, and setting times of the pastes with varying alkali contents seem to be contradictory with the resistivity development, (i) while the terminal values of the resistivity curves of specimens with increasing alkali content failed steadily (Figure 2, except for SL6% and SL7%, of which values were very similar), the fluidity and shear viscosity of the fresh pastes (Figure 5 and Figure 6), which certainly affected the ions transmission and electrical conductivity, only increased dramatically when alkali content surpassed 6 wt%. (ii) The setting time, another index for judging the maturity of hydration products, increased steadily before the alkali content reached 7 wt% and then suddenly increased at 8 wt% (Figure 7). One may be confused by the prolonged setting time since the increased activator content should have accelerated the dissolution of the slag particles, which lead to a faster setting, as presented in some studies [23,24], wherein the modulus of the water glass was relatively low (typically lower than 1.25). However, for those studies using high-modulus water glass (typically higher than 1.6), increasing the alkali content tended to prolong the setting time [25,26], which was probably due to a large amount of precipitation forming in the pore solution and the surface of the slag quickly inhibiting the dissolution and diffusion control in dominance. These facts showed a negative effect of increasing the alkali content to a very high value (8 wt%); that is, decreasing the viscosity and, therefore, mitigating the further hydration. More discussions regarding calorimetry (Section 3.2) and microscale characterizations (Section 3.3) are necessary to explain the range between 4 wt~7 wt%, and to rationally make quantitative relationships between resistivity and setting/strength development (Section 3.4 and Section 3.5), by which application of resistivity in non-destructive monitoring would be more convincible.

### 3.2. Reaction Heat Analysis of the Slag-Based Geopolymer with Different Alkali Contents

Generally, two peaks were identified on the heat release rate curves (Figure 8), in which the range before point A of the initial heat release peak was due to the initial dissolution and drastic reaction on the surface of the particles, corresponding to stage I, which was classified by the resistivity curves (Figure 3), and range AB was attributed to stage II. The second heat release peak was relatively vague in identifying the reaction stages in detail. It comprised the induction period, acceleration, and deceleration periods.

The heat release rate supported the above explanation for the variations of resistivity curves with varying alkali contents. For the initial peak, an increase in alkali content shortened the time before reaching point A (1.6 h for SL4% and 50 min for SL8%). This was due to the increase in alkali content leading to an increase in OH^−^ concentration in the solution, accelerating the dissolution rate of slag particles. Consequently, there was more precipitation between slag particles per unit of time (see the enlarged region in Figure 8). More specifically, the initial peak width of sample SL8% was obviously narrower than the others, again showing that the reaction was inhibited.

Except for sample SL4%, the other samples exhibited the second heat release peak. It seems that the inhibition effect of increasing alkali content induced such peaks, but their position and intensity were not proportional to the alkali content. While the position was advanced for sample SL (5 wt~8 wt%), the intensity fluctuated; it increased dramatically for sample SL5% and then decreased when alkali content reached 6 wt%, followed by a re-increase as the alkali content kept increasing to 8 wt%. The fluctuation was due to the two-fold effect of increasing alkali content as also depicted by the resistivity curves (Figure 2) and workability (Figure 5 and Figure 6): (i) promote the local reactant concentration and (ii) decrease the fluidity. From these facts, an intense acceleration peak on the heat release rate curve (typically for sample SL8%) did not necessarily lead to a higher global reaction degree and higher strength, as the reaction may be uneven, which was supported by the increasing setting time for sample SL8% compared with that of sample SL (4 wt~7 wt%). Certainly, there existed a threshold for adjusting the alkali content, and by speculation, it may be 6 wt%, since there were inflections in the fluidity curve and shear viscosity curve at this value (Figure 5 and Figure 6).

### 3.3. Microstructural Evolutions of the SL Pastes with Different Alkali Contents

#### 3.3.1. FTIR Analyses

In order to further verify the difference of reaction degree detected by resistivity, FTIR tests were carried out on the specimens cured for 4 h, 24 h, and 7 days to show the evolution of the functional group (i.e., Si–O–T, where T refers to Si and Al) in the slag-based geopolymer pastes (Figure 9).

For the activating products of a high Ca-system, such as slag, the layered C–A–S–H gel is the major reaction product [11,27,28,29], wherein the Si–O–T bonds are the major linkages. The bands that appeared at ~3548, ~3475, and ~3413 cm^−1^ could be attributed to the asymmetric stretching vibration of the –Si–OH bond. The intensities of these bands seemed to increase with time, representing a transformation from the intermediate (or immature) complex to the more developed product. In the C-A-S-H system, the stretching vibration of the Si–O–T (T = Al, Si) bond is mainly located within the range between 950~1000 cm^−1^ [30,31,32]. As the reaction progressed from 4 h to 24 h, the Si–O–T bond shifted to a lower frequency, namely from 995.09 to 987.37 cm^−1^. The Si–O–T vibration band was thought to be determined by the Al atom content per unit of formula. As the content of tetrahedral Al atom increased in the aluminosilicate chain, the central band of Si–O–T moves to a lower frequency due to the longer bond length and smaller bond angle of Al–O–Si compared with that of Si–O–Si [30]. On the other hand, there was a trend of an increasing wavenumber of the Si–O–T band when decreasing the alkali content (Figure 9a), with a small discrepancy of 5 to 10 cm^−1^, which suggested a slight increase in the polymerization degree of the C–A–S–H structure. As the reaction progressed from 24 h to 7 days, the Si–O–T bond returned to higher frequencies. This behavior could be related to the slag reaction process, during which more Si–O groups in the original slag were broken, raising the Si concentration in the pore solution and reconstructing the gelling phase. The product transformed into the Si-enriched state [30,33]. In summary, a smaller alkali content led to a higher degree of geopolymerization.

#### 3.3.2. SEM-EDS Results

Figure 10 illustrates the effect of alkali content on the micro-morphology of the slag-based geopolymer pastes cured for 7 days. When the alkali content was within the range of 4–5 wt%, an increase in alkali content led to a denser hardened interface, as well as a decreased amount of crack, indicating a higher polymerization degree. When the alkali content continued to increase from 5 to 8 wt%, the hardened interface of the paste was relatively rough, where many cracks could be observed. Many debris particles (circled in Figure 10) were also embedded in the matrix, which mitigated the strength of the paste, as presented in the following section.

To further determine the composition of a slag-based geopolymer with different alkali contents, EDS analyses were carried out (Table 2). For a slag-based geopolymer, the Ca/Si ratio of the hardening products was generally approximately 1.0 [6]. In the current study, those samples with a lower alkali content (4~5 wt%) exhibited a higher Ca percentage (see points 1~4 in Table 2) and a higher Ca/Si ratio (~1.4). This was due to the increasing alkali content preventing the dissolution of slag particles by reducing the liquid water and mitigating the ions transmission, as depicted by the model in Figure 4. The SEM and EDS results reflected a general trend of higher calcium and silicon ratio when the alkali content was 4–5 wt%.

When the alkali content was above 5 wt% (that is, an over-high content of alkali), the EDS results of the points in Figure 10c–e show that the Ca/Si ratio of the product decreased. Higher alkali content led to a lower Ca/Si ratio in the product. There was an optimal value of sodium silicate. When the proportion of sodium silicate (or to say the alkali content) reached 6 wt%, the slag particles became isolated from the liquid due to the extra reaction product (depicted in Figure 4) and lower fluidity [11]. In addition, a carbonation reaction occurred easily when excessive sodium silicate was added [34]. The EDS results of points 7 and 10 indicate the likely formation of carbonization products. At 6–8 wt% alkali content, an increase in alkali content led to an increase in silicon content in the product, whereas the Ca/Si in the pastes decreased to around 0.5 when the alkali content reached 7–8 wt%. In addition, the increase in alkali content increased the probability of carbonation reaction [34], thus influencing the development of strength, as will be shown in Section 3.4.

### 3.4. The Effect of Alkali Content on Compressive Strength and Flexural Strength of the Slag-Based Geopolymer

To further investigate the relationship between resistivity and early-age performance of a slag-based geopolymer, compressive and flexural strength tests were carried out on the slag-based geopolymer pastes at different ages (7, 14, and 28 days, see Figure 11). In the time dimension, both the compressive strength and the flexural strength increased with increasing curing age. This was not surprising, but there appeared to be a threshold in terms of alkali content, as also reflected by the preceding characterizations. The strength increased when increased the alkali content from 4 wt% to 5 wt%, whereas the strength fell obviously when the alkali content kept increasing to 8 wt%.

The strength development again presented the two-fold effect of increasing alkali content on the hydration of slag-based geopolymer paste, as had been described in Section 3.1.2. When the alkali content varied in an appropriate range (i.e., 4~5 wt%), increasing the alkali content effectively accelerated the early-age reaction and denser the microstructure (proved by the SEM results, see Figure 10a,b). When the alkali content surpassed the optimal value (i.e., 6 wt%), both the compressive strength and flexural strength declined, accompanied by dramatic diminishments in the viscosity and fluidity (Figure 5 and Figure 6). From a microscale perspective, the excessive amount of sodium silicate (i) raised the pH value; (ii) decreased the liquid water content; and (iii) increased the Si content and caused quick precipitation of C–S–H gel in the pore solution, as well as on the surface of the slag particles [35,36,37,38]. The three factors trapped the ionic transmission, especially the dissolution of Ca^2+^ ions. This further separated the aluminosilicate source and alkaline activators (as also depicted by the model in Figure 4), resulting in a less-developed three-dimensional network [34]. In addition, water evaporation was affected by the alkali content during the condensation reaction [39]. Excessive Na^+^ may form sodium carbonate when it comes into contact with CO_2_ from the atmosphere, reducing the strength value. The measured Ca/Si ratio in Table 2 also verified the differences of hydration products formed in samples with varying alkali contents. Those samples with a higher compressive strength had a higher Ca/Si ratio, which could explain the lower 7-day compressive strength of a slag-based geopolymer with a very high alkali content (8 wt%).

### 3.5. Could Resistivity Be Linked to the Strength Development and the Setting Behavior?

In previous research, higher resistivity denotes a higher strength value [15]. In the current study, the sample with an alkali content of 5 wt% achieved the best mechanical performance, but the resistivity curve did not reflect this trend. As such, it was unable to directly figure out a quantitative relationship between resistivity and strength value. However, there was a vague tendency. It could be found in Figure 2 that the later-age resistivity value (see the terminal of the curves) decreased as the alkali content increased. Such a trend was similar to that of the strength development except for sample SL5%, of which workability was very similar to that of sample SL4%. That is, when increasing the alkali content of a slag-based geopolymer, if the workability is nearly unchanged, a decreased resistivity may denote a higher strength value; if the workability is diminished, then the decreased resistivity brings about a lower strength value. This may be applicable in engineering cases.

Figure 7 showed that the initial setting time increased in line with the alkali content. Some studies [32,33,34] have found that initial and final setting times decrease as the alkali content increases; however, the results of other studies were similar to those in this research [35,36,37], potentially due to the low amount of water in a slag-based geopolymer with a high alkali content, which renders the brief hydrolysis of sodium silicate insufficient. Consequently, the release of active monomers in the raw materials can be delayed, thereby increasing the setting time.

In the hardening process of a slag-based geopolymer, the characteristic point A/B on resistivity curves can be obtained in Figure 2 and Figure 3. As shown in Figure 12 and Figure 13, the relationships between the initial/final setting time and points A/B were nonlinear. The relationship between the initial setting time and point A was as follows:*y* = 2284.24 + 62267.44 × *exp* (−*x*/427.34)(1)
where the correlation coefficient *R*^2^ = 0.98214.

Point B represents the maximum polymerization rate, and the resistivity growth rate reaches the maximum, as shown in Figure 3. By fitting the relationship between point B and the final setting time, the following formula can be obtained:*y* = 19480.29 − 39439.13 × *exp*(−*x*/1332)(2)
where the correlation coefficient *R*^2^ = 0.99399.

Overall, points A/B of the resistivity curve and the initial/final setting time of a slag-based geopolymer were related. The initial setting time and final setting time of the slag-based geopolymer with different alkali contents can be deduced from the characteristic point on the electrodeless resistivity curve. A similar correlation was found between the resistivity and setting time in the hardening process of cement-based materials [40], implying that this method is suitable for predicting or determining the early performance of cement-based materials. The result obtained in this research also exhibited a similar correlation.

## 4. Conclusions

This study investigated the influence of increasing alkali content on the early-age hydration process of slag-based geopolymer paste. Upon recording the development of resistivity, workability, strength, and micro-morphology, the two-fold effect of increasing alkali content, as well as the applicability of resistivity in monitoring the hydration process, were discussed. The following conclusions can be drawn:(1)As a relatively novel characterization, electrodeless resistivity measurement showed that although increasing the alkali content accelerated the dissolution stage of the slag-based geopolymer paste, the reaction seems to be inhibited, as presented by the differential curve of resistivity. However, such an inhibition effect could not be reflected by the heat release rate curve, showing the superiority of resistivity of in situ monitoring of the hydration process of a slag-based geopolymer.(2)The setting time, viscosity, micro-morphology, and strength development showed further explanation of the influence of alkali content. When the alkali content ranged from 4 to 5 wt%, the viscosity and setting behavior were nearly unchanged, while the strength increased dramatically, especially the 7 days compressive strength (the amplification was about 10%). When the alkali content surpassed 6 wt%, the viscosity was decreased, accompanied by the prolonged setting time and decreased strength value. This supplied the result of resistivity. That is, a high level of alkali content may lead to heterogeneous hydration of a slag-based geopolymer, and the reaction system was dominated by diffusion control.(3)There existed a quantitative relationship between resistivity and initial/final setting time. However, the former could not be quantitatively correlated with the strength value, which was unlike the cases in previous studies where the alkali content was kept unchanged.

## Figures and Tables

**Figure 1 polymers-15-00518-f001:**
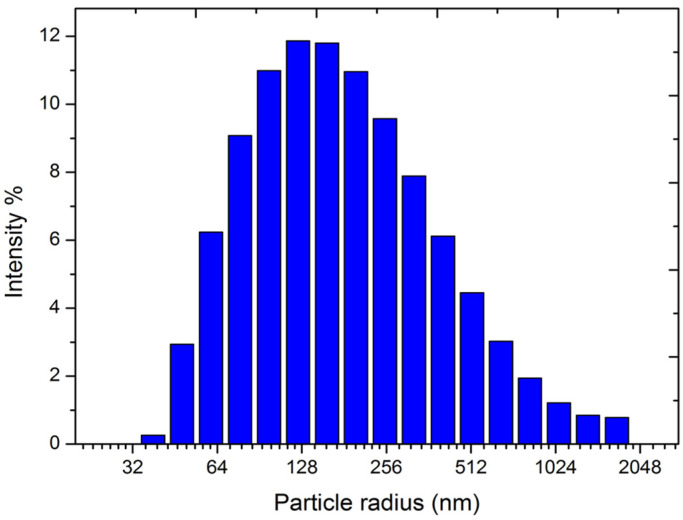
Particle size distribution of BFS.

**Figure 4 polymers-15-00518-f004:**
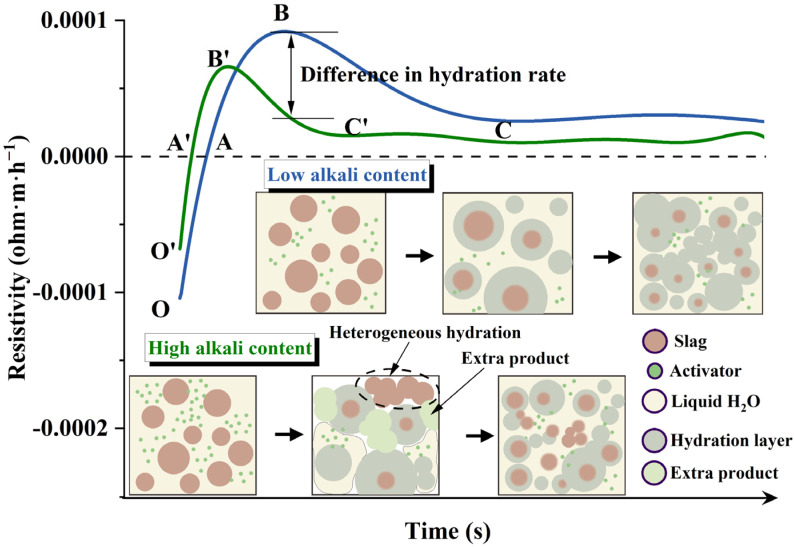
A simplified model of the effect of alkali content on resistivity development.

**Figure 5 polymers-15-00518-f005:**
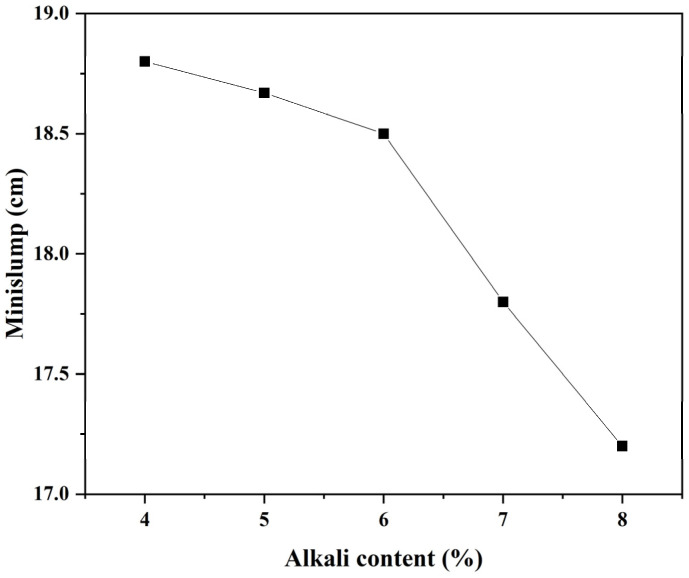
Effect of alkali content on fluidity.

**Figure 6 polymers-15-00518-f006:**
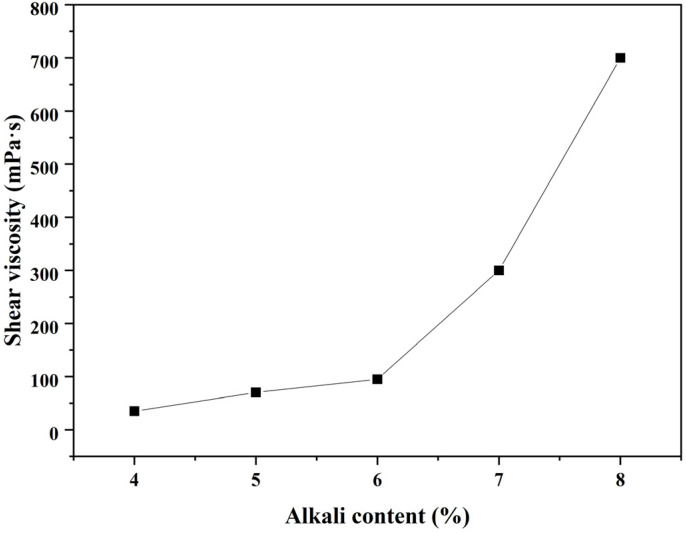
Effect of alkali content on shear viscosity.

**Figure 7 polymers-15-00518-f007:**
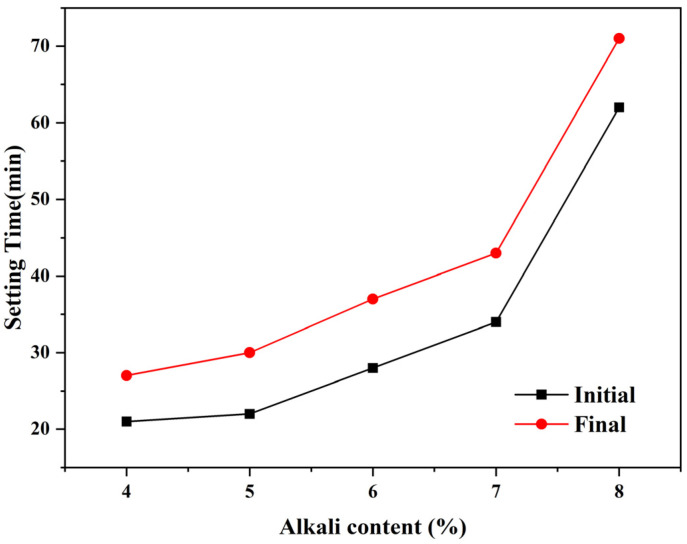
Effect of alkali content on setting time.

**Figure 8 polymers-15-00518-f008:**
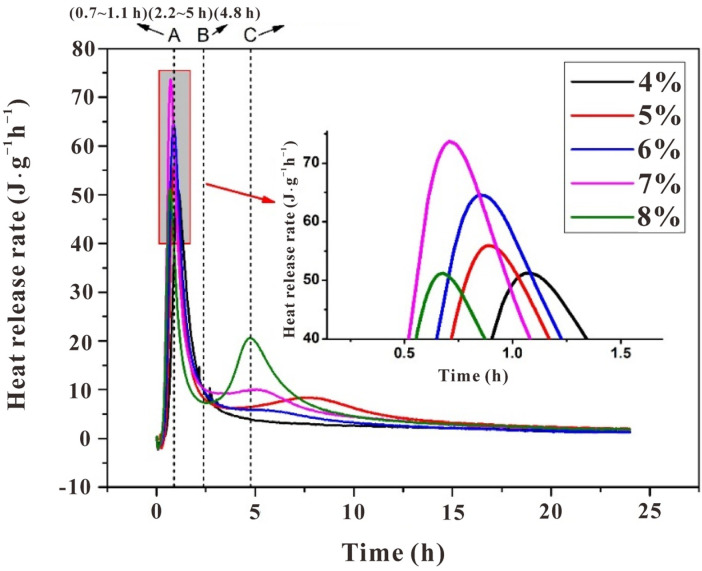
Effect of alkali content on heat release of an early-age slag-based geopolymer.

**Figure 9 polymers-15-00518-f009:**
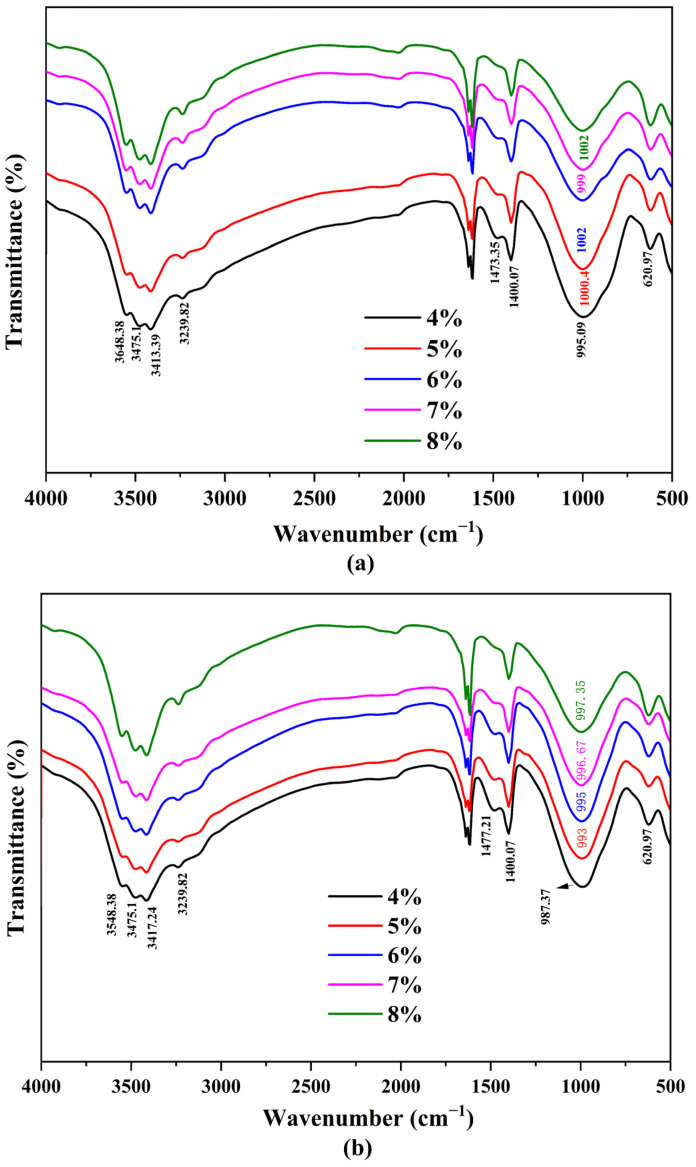
FTIR spectra of the slag-based geopolymer paste with different alkali contents at the age of (**a**) 4 h, (**b**) 24 h, and (**c**) 7 days.

**Figure 10 polymers-15-00518-f010:**
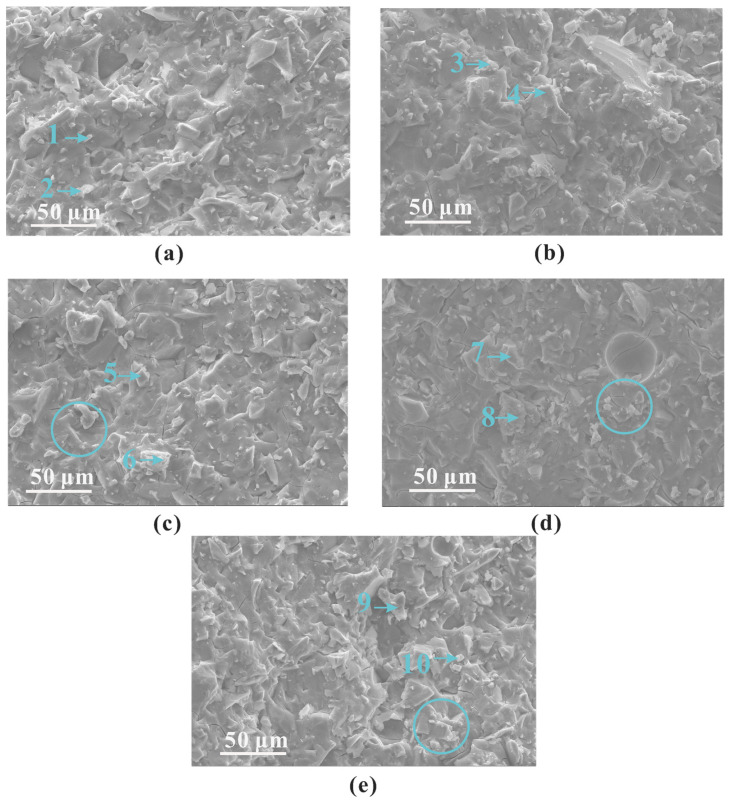
SEM results of slag-based geopolymer pastes cured for 7 d: (**a**) SL4%; (**b**) SL5%; (**c**) SL6%; (**d**) SL7%; and (**e**) SL8%. Note that the numbers in the images refer to the EDS numbers in Table 2.

**Figure 11 polymers-15-00518-f011:**
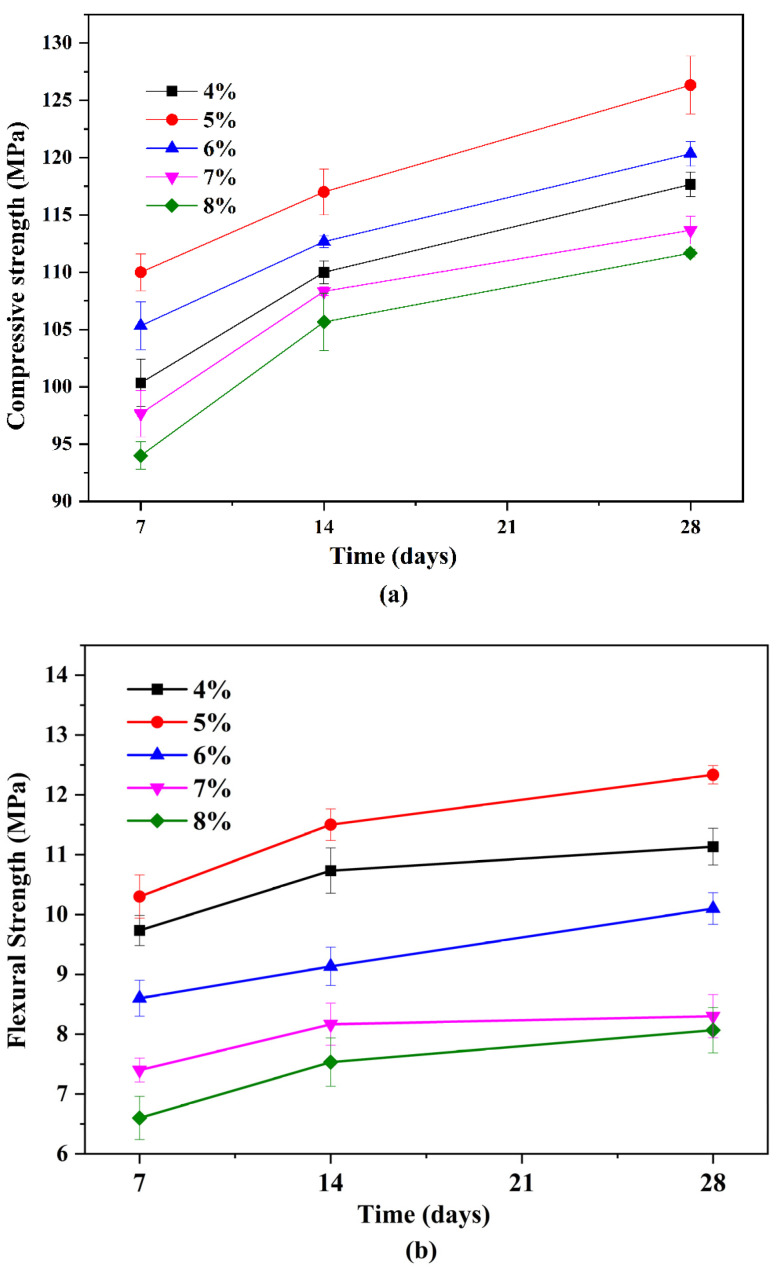
Relationship between the alkali content and (**a**) compressive strength and (**b**) flexural strength.

**Figure 12 polymers-15-00518-f012:**
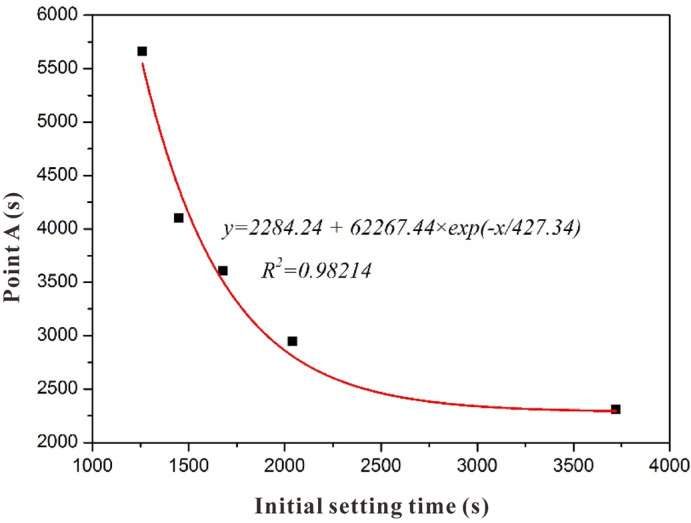
Relationship between characteristic point A of the resistivity and initial setting time.

**Figure 13 polymers-15-00518-f013:**
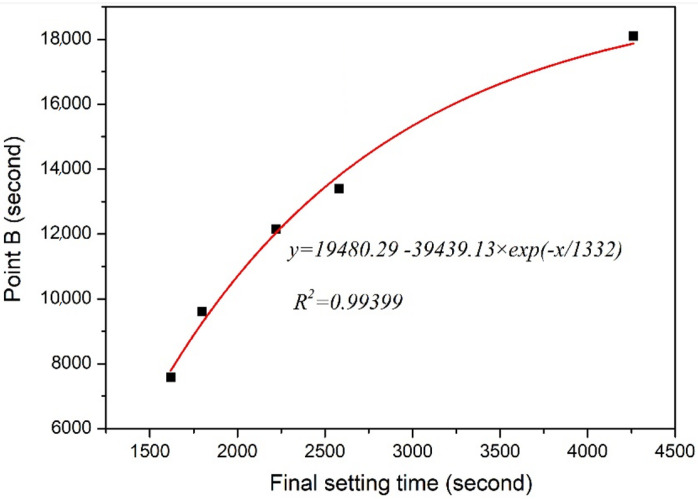
Relationship between characteristic point B of the resistivity and final setting time.

**Table 1 polymers-15-00518-t001:** Chemical components of the BFS (mass fraction%).

Composition	CaO	SiO_2_	Al_2_O_3_	MgO	SO_3_	TiO_2_	K_2_O	Fe_2_O_3_	MnO	Na_2_O	L. O. I
Content (wt%)	41.5	32.6	14.7	6.48	2.48	0.58	0.394	0.349	0.29	0.25	0.38

**Table 2 polymers-15-00518-t002:** Chemical element compositions of products tested by EDS.

Alkali Content		Element	Atomic Percentage (wt%)							Ca/Si
	Position		C	O	Na	Mg	Al	Si	Ca	
4 wt%	1		7.69	60.28	1.57	3.81	5.37	8.79	12.50	1.4
4 wt%	2		9.71	58.93	4.96	2.77	4.13	9.35	10.10	1.08
5 wt%	3		5.92	58.82	0.21	3.74	6.21	10.71	14.39	1.34
5 wt%	4		9.68	63.65	2.88	3.71	4.04	8.10	10.93	1.35
6 wt%	5		11.94	57.46	7.31	2.01	3.48	9.57	8.23	0.86
6 wt%	6		6.59	51.39	3.97	1.45	3.65	19.95	14.99	0.75
7 wt%	7		7.29	60.23	10.87	1.39	2.70	11.02	6.52	0.59
7 wt%	8		11.14	64.75	2.72	1.63	4.99	8.08	4.69	0.58
8 wt%	9		7.37	60.31	11.32	1.72	2.80	11.07	5.40	0.49
8 wt%	10		7.41	57.13	11.38	1.38	2.89	12.83	6.98	0.54

## Data Availability

The data presented in this study are available in article.

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
