# Peer review of "The Influence of Alkali Content on the Hydration of the Slag-Based Geopolymer: Relationships between Resistivity, Setting, and Strength Development"

_polymers, 2023, doi:10.3390/polym15030518_

Round 1

Reviewer 1 Report

The topic evaluated by these Authors is certainly interesting and worthy of investigation, but the format of the manuscript needs extreme care, at the present form it seems it is a collage of different pieces taken from various different studies. Other suggestions are reported in the attached .pdf aim at improving the quality of the paper. The results, anyway, are worthy of investigation after a minor revision.

Reviewer 2 Report

It is interesting to read the manuscript describing the alkali content on the hydration of slag-based geopolymer. However, the results of this manuscript should be verified and confirmed. I suggest that the text of the article at this stage is not up to the standard of Polymers publication. It should have a major version and the suggestions are as follows:

- The abstract should state the important findings of the research.

- The authors should have clearly stated the critical points in the introduction section. In particular, the addition of alkali activators to slag should be classified as either a geopolymer or an alkali activated material.

- The authors should state the highlights of this study, as many studies have been conducted on this topic in the past.

- The author should explain the mixing procedure of the specimens, including the mixing process of the alkali activators.

- The authors should have further clarified the calculation of the concentration of alkaline activators (%).

- The authors should have specified how many specimens were used for each test.

- Figure 7: The higher the percentage of alkali activators, the longer the setting time, which seemed to be contrary to the general findings. The authors should have stated such results and cited the relevant literature to support the findings.

- The trend of compressive and flexural strength did not seem to increase significantly due to the increase in the concentration of alkali activators, and the authors would like to further explain this mechanism.

- The results of the tests should be compared with the results of relevant studies and supported by appropriate citations of relevant literature.

- Improvement in each section of the results is necessary. Each section also needs a better comparison with the research topic and relevant references.

- The conclusion should address the main findings of the study.

Round 2

Reviewer 2 Report

The author gave a more detailed response to the comments, which means that the author has done a lot of work to perfect this paper. I think this manuscript can be accepted in its present state in Polymers.